

# Impact of phase state and non-ideal mixing on equilibration timescales of secondary organic aerosol partitioning

Meredith Schervish[1] and Manabu Shiraiwa[1]

[1]Department of Chemistry, University of California, Irvine, CA 92697, USA

**Correspondence:** Manabu Shiraiwa (m.shiraiwa@uci.edu)

**Abstract.** Evidence has accumulated that secondary organic aerosols (SOA) exhibit complex morphologies with multiple phases that can adopt amorphous semisolid or glassy phase states. However, experimental analysis and numerical modeling on formation and evolution of SOA still often employ equilibrium partitioning with an ideal mixing assumption in the particle phase. Here we apply the kinetic multilayer model of gas-particle partitioning (KM-GAP) to simulate condensation of semi-
volatile species into a core-shell phase-separated particle to evaluate equilibration timescales of SOA partitioning. By varying bulk diffusivity and activity coefficient of the condensing species in the shell, we probe the complex interplay of mass transfer kinetics and thermodynamics of partitioning. We found that the interplay of non-ideality and phase state can impact SOA partitioning kinetics significantly. The effect of non-ideality on SOA partitioning is slight for liquid particles, but becomes prominent in semi-solid or solid particles. If the condensing species is miscible with low activity coefficient in the viscous
shell phase, the particle can reach equilibrium with the gas phase long before the dissolution of concentration gradients in the particle bulk. For the condensation of immiscible species with high activity coefficient in the semisolid shell, the mass concentration in the shell may become higher or overshoot its equilibrium concentration due to slow bulk diffusion through the viscous shell for excess mass to be transferred to the core phase. Equilibration timescales are shorter for the condensation of lower volatility species into semisolid shell; as the volatility increases, re-evaporation becomes significant as desorption is
faster for volatile species than bulk diffusion in semisolid matrix, leading to an increase in equilibration timescale. We also show that equilibration timescale is longer in an open system relative to a closed system especially for partitioning of miscible species; hence, caution should be taken when interpreting and extrapolating closed system chamber experimental results to atmosphere conditions. Our results may reconcile apparent discrepancies between experimental observations of fast particle-particle mixing and predictions of long mixing timescales in viscous particles and provide useful insights into description and
treatment of SOA in aerosol models.

## 1   Introduction

Secondary organic aerosols (SOA) are ubiquitous in the atmosphere, comprising a major fraction of atmospheric aerosols (Jimenez et al., 2009). They have impacts on air quality, public health and climate, and their representation in large-scale models remains a major challenge in atmospheric chemistry (Oak et al., 2022; Shrivastava et al., 2017a). SOA is generated through
oxidation of volatile organic compounds and condensation of semi- and low-volatility products into pre-existing particles. This





involves a series of transport processes including gas-phase diffusion, gas-surface transfer via reversible adsorption, surface-bulk exchange, and bulk diffusion within the particle (Shiraiwa et al., 2014). These processes are often represented as very fast relative to the timescales of other atmospheric processes associated with SOA, so that the gas and particle are traditionally regarded as being in instantaneous equilibrium.

There has been mounting evidence that SOA particles can exist in amorphous semi-solid or glassy states depending on chemical composition, relative humidity and temperature (Zobrist et al., 2008; Virtanen et al., 2010; Renbaum-Wolff et al., 2013; Reid et al., 2018). Global and regional modeling studies have shown that organic aerosols exist in an amorphous solid or glassy state (Rasool et al., 2021; Shiraiwa et al., 2017; Li et al., 2021; Schmedding et al., 2020), facilitating long-range transport of organic compounds in the atmosphere (Mu et al., 2018; Shrivastava et al., 2017b). The occurrence of a highly viscous state

can lead to kinetic limitations of bulk diffusion within the particle to retard heterogeneous reactions and evaporation kinetics (Zhou et al., 2013; Perraud et al., 2012; Vaden et al., 2011; Shiraiwa et al., 2011; Lam et al., 2019). Uptake coefficients of isoprene epoxydiols (IEPOX) are observed to be reduced by organic coatings, affecting particle morphology and chemical evolution of SOA (Zhang et al., 2018; Lei et al., 2022; Olson et al., 2019). Modeling work also demonstrates the impact of diffusion limitations in highly viscous aerosol on partitioning of semi-volatile compounds thus affecting evolution of particle

size distribution (Shrivastava et al., 2022; Cummings et al., 2020; Shiraiwa et al., 2013a, b; Zaveri et al., 2022, 2020; He et al., 2021),.

       Equilibration timescales can be prolonged in highly viscous particles under low relative humidities and temperatures (Li and Shiraiwa, 2019; Shiraiwa and Seinfeld, 2012). In fact, a recent field observation in Beijing demonstrated that partitioning of organic species was retarded to limit SOA mass when ambient particles adopt a semi-solid state (Gkatzelis et al., 2021).

Chamber experiments have observed limited particle-particle mixing of SOA , while some experimental studies indicate little kinetic limitations and fast mixing between miscible organic species (Ye et al., 2018; Loza et al., 2013; Ye et al., 2016; Robinson et al., 2013; Saleh et al., 2013; Habib and Donahue, 2022). These contrasting results leave questions open about under what conditions and properties can kinetically-limited or equilibrium gas-particle partitioning be expected in the atmosphere (Mai et al., 2015).

Atmospheric aerosol particles are generally complex mixtures of organics and inorganics, and water (Riemer et al., 2019). Such mixtures may exhibit non-ideal behavior possibly with phase separation when the organic components have low O:C ratio and often exist in core-shell structures (Song et al., 2012; Freedman, 2020; You et al., 2014; Zuend and Seinfeld, 2012). Aerosols composed of two organic populations may also phase separate into two phases (usually a highly oxidized hydrophilic phase and a low O:C hydrophobic phase) (Freedman, 2017; Gorkowski et al., 2020; Song et al., 2017; Pöhlker et al., 2012).

Very recently, Huang et al (2021) even showed that the particles consisting of primary organic compounds, SOA, and inorganic compounds can phase separate into three phases (Huang et al., 2021).

       Many studies still assume a homogeneous well-mixed particle phase as well as ideal mixing between the particle and condensing gas-phase species. Because a dynamic consideration of gas-particle equilibrium would be significantly more computationally expensive especially in larger scale models, it is important to understand under what conditions this consideration is

most critical (Mai et al., 2015; O'Meara et al., 2016). Based on current experimental techniques, it is challenging to determine





whether any limitations to SOA partitioning or mixing are due to diffusion limitations, small mass accommodation coefficients, or non-ideality and low miscibility between the populations studied. In this study, we investigate the combined effects of particle phase state and non-ideal mixing by modeling condensation of semi-volatile species into core-shell particles to evaluate equilibration timescales of SOA partitioning.

## 2  Model Description

The kinetic multi-layer model of gas-particle interactions in aerosols and clouds (KM-GAP) has been described in detail previously (Shiraiwa et al., 2012, 2013b). A brief description and the specifics of the implementation for this work are addressed here. We assume a particle of a core-shell morphology, mimicking the shell phase with predominantly organics and the core phase as aqueous inorganics. KM-GAP consists of a gas-phase, a sorption layer, a near-surface bulk layer (nsb), and a number

of bulk layers. As shown in Fig. 1, we represent the shell phase with a near-surface bulk layer and four bulk layers, while the core is represented with one layer. Sensitivity studies were conducted with more layers (10 and 20 layers) to ensure that the simulation results were not sensitive to the number of layers chosen, but it only affects resolution of bulk concentration profiles. KM-GAP explicitly treats the processes of gas-phase diffusion, adsorption and desorption, surface-to-bulk transport, as well as bulk diffusion within the shell and transport between the shell and core phases. A monodisperse particle population with

an initial particle diameter of 200 nm and total particle concentration of $10^4$ cm$^{-3}$ is assumed, which yields a total organic aerosol mass of $\sim$40 $\mu$g m$^{-3}$. The core and shell are composed of different non-volatile species each with a molar mass of 200 g mol$^{-1}$. Initially, the total amount of mass in the core and shell are the same so that there will be an equal amount in each phase at equilibrium when the activity coefficients of the condensing species in 1 in both phases.

A gas-phase species is present with the same molar mass and density as the particle phase species, but with pure compound

saturation mass concentration of $C^0 = 10$ $\mu$g m$^{-3}$. Volatility or effective saturation mass concentration ($C^*$) is expressed as $C^* = \gamma_{\text{shell}}C^0$, which controls the partitioning by driving the surface to bulk transport in the model (Donahue et al., 2006). The initial gas-phase concentration of the condensing species is set to be small, $10^{-7}\mu$g m$^{-3}$, so that condensation of trace amounts of condensing species would not alter particle diameter and properties including phase state and non-ideality. It also ensures that the approximate volume of each layer is maintained even in circumstances with slow diffusion. We mainly consider a

closed system, in which condensation of species would lead to a decrease in its gas-phase mass concentration and an increase in its particle-phase mass concentration, while we also conduct additional simulations with an open system with a fixed gas-phase concentration. The surface accommodation coefficient is set to be 1 based on previous measurements and molecular dynamics simulations (Riipinen et al., 2011; Julin et al., 2014; von Domaros et al., 2020).

The bulk diffusivity ($D_{\text{b,shell}}$) is varied in the range of $10^{-5}$ to $10^{-20}$ cm$^2$ s$^{-1}$ covering the range of particle being in a

liquid, semi-solid to amorphous solid or glassy phase state (Shiraiwa et al., 2011). $D_{\text{b}}$ is fixed at any given depth in the particle bulk in each simulation, assuming that condensation would not alter particle viscosity and diffusivity, as only trace amounts of species condense to preexisting particles in these simulations. The activity coefficient of the condensing species in the shell is varied in the range of $10^{-3}$ to $10^5$ covering and expanding the range of activity coefficients measured or modeled in between



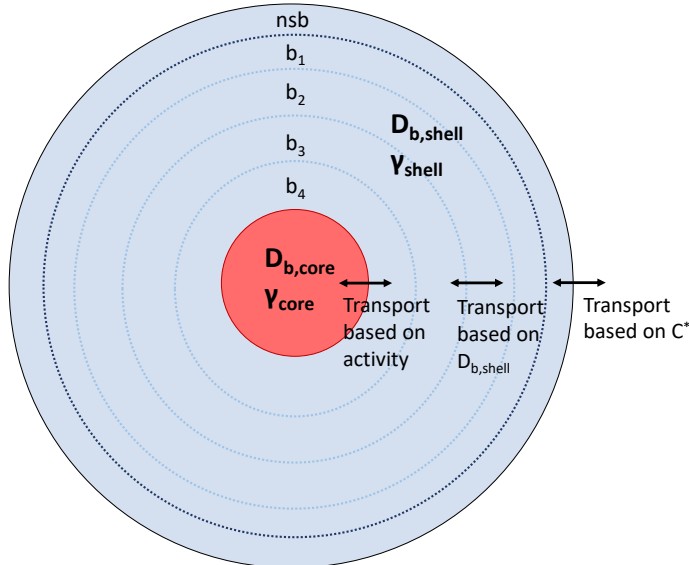

**Figure 1.** KM-GAP model representation showing the different particle layers with a near-surface bulk layer (nsb) and 4 additional bulk layers (b1-4). In this work $D_{b,core} = 10^{-8}$ and $\gamma_{core} = 1$ for all runs while $D_{b,shell}$ is varied between $10^{-5}$ to $10^{-20}$ cm$^2$ s$^{-1}$ and $\gamma_{shell}$ is varied in the range $10^{-3}$ to $10^5$.

atmospherically relevant compounds (Liu et al., 2020; Chang and Pankow, 2006; Zuend et al., 2008; Shiraiwa et al., 2013b). The activity coefficient of the condensing species in the core is fixed to be 1 for all simulations. The e-folding equilibration timescale, $\tau_i$, is calculated the first time (t) the following inequality is satisfied.

$$\frac{|C_i(t) - C_{i,\text{eq}}|}{|C_{i,0} - C_{i,\text{eq}}|} < \frac{1}{e} \tag{1}$$

where $C_i$, is mass concentration of the condensing species in the phase $i$ (shell, core, or total particle) at e-folding equilibration time ($\tau$), initial time (0), and equilibrium (eq).

## 3 Results and discussion

We first show the simulation results of a few representative cases to illustrate how partitioning and mass transport of the condensing species in the semisolid shell of a phase-separated particle is affected by non-ideality. Then we discuss the overall combined effects of non-ideality and particle phase state on equilibration timescales of SOA partitioning.

### 3.1 Temporal evolution of SOA partitioning

Fig. 2 shows the simulation results for condensation of semi-volatile species ($C^0 = 10$ $\mu$g m$^{-3}$) into core-shell particles: the shell is semi-solid with $D_{b,shell} = 10^{-16}$ cm$^2$ s$^{-1}$ and the core is liquid with $D_{b,core} = 10^{-8}$ cm$^2$ s$^{-1}$. The activity coefficient





of the condensing species in the shell phase ($\gamma_{\text{shell}}$) is set to be (a) 1, (b) $10^{-3}$, and (c) $10^5$, while that in the core phase is set to be 1 for all cases. Through these representative cases, the effects of non-ideality and phase state on timescales of equilibration can be realized.

In Fig. 2, the x-axis is simulation time in seconds and the y-axis gives the number concentration of the condensing species in molecules per volume. The blue lines show the concentration in each shell layer, getting progressively lighter in color from the near-surface bulk layer to the innermost shell layer (nsb, b1-4). The average concentrations in the shell (black), core (red), and whole particle (dashed green) are also shown. These concentrations increase over time as condensation of semi-volatile species progresses. In some cases, these concentrations need to decrease to reach their equilibrium values due to interplay of slow diffusion and non-ideality as discussed below.

The timescales, calculated from Eq. 1 for the shell ($\tau_{\text{eq,shell}}$), core ($\tau_{\text{eq,core}}$), and total particle ($\tau_{\text{eq,tot}}$) can be identified and are shown on the plots as the vertical lines. In the two non-ideal cases, $\tau_{\text{eq,tot}}$ is equivalent to one of the other timescales and thus a single line is labeled with both. It is useful to consider the total amount of the condensable vapor that can be being held by each phase within the particle, which is a function of the total mass concentration, the pure compound saturation mass concentration, and the activity coefficient in each phase. Because the mass of each particle phase is practically the same, the fraction of the condensing species that will end up in the core relative to the shell would be the ratio of the inverse of the activity coefficients, which is equal to the activity coefficient in the shell as the activity coefficient in the core is set to be 1. Hence, at high $\gamma_{\text{shell}}$, the core will retain most of the mass and $\tau_{\text{eq,tot}}$ is controlled by $\tau_{\text{eq,core}}$, while $\tau_{\text{eq,tot}}$ is similar to $\tau_{\text{eq,shell}}$ at low $\gamma_{\text{shell}}$ with the shell containing most of the condensing mass at equilibrium.

In all three cases shown here, the mass concentrations in the shell may become higher than its equilibrium concentration due to slow bulk diffusion through the shell for excess mass to be transferred to the core phase. This overshooting behavior is most prominent in the case where the semi-volatile species is immiscible with the shell phase (e.g., $\gamma_{\text{shell}} = 10^5$) as concentrations in all shell layers overshoot their equilibrium values. This behavior also leads to an interesting consideration of what the equilibration timescale represents, as this will lead to two different times when the inequality in Eq. 1 holds. For example, in Fig. 2c, the first time this equality is met is at $t = 2$ ms followed by a period where the concentration in the shell is outside the inequality because it is too high. This condition is once again satisfied at $t = 2.5$ hr. The first one corresponds to the time when the average concentration in the shell is close to its steady state value, while the second value is representative of when the concentration gradients within the shell are close to dissolving to reach full equilibrium within the particle. Note that there are some cases where this same overshoot occurs, but the extent of overshooting is so small that the inequality in equation 1 holds throughout the whole range. Realistically if there is such a small change in the concentration, the equilibrium can likely be considered to be holding throughout the whole range. This occurs in Fig 2b where the equilibration timescale for the shell with the gas phase is found to be roughly 7 minutes, while there is still concentration gradient and the shell has not reached full equilibrium yet. The timescale for the concentration gradients to dissipate closely follows the equilibration timescale of the core. Therefore, $\tau_{\text{eq,core}}$ should be used to discuss the relaxation timescale of concentration gradients in the shell. This overshooting phenomenon is only seen in the shell and both values will be discussed here as each one may be more relevant in different circumstances. For example, in the case of acid-catalyzed IEPOX reactive uptake, it may be important to know





the concentration of IEPOX in the innermost region of a core-shell particle with an aqueous core as this would be where the reaction occurs (Zhang et al., 2018).

In the ideal case ($\gamma_{\text{shell}} = 1$), the shell reaches a steady state within a few minutes as the shell can hold a substantial amount
of the condensing material. A concentration gradient is maintained through the shell due to kinetic limitations of bulk diffusion though the shell to the core, resulting in $\tau_{\text{eq,core}}$ to be about 4 hours. Despite diffusion in the shell being slow with a timescale of a couple hours for relaxation of concentration gradients, the condensing species reaches its equilibrium concentration in the particle significantly faster with $\tau_{\text{eq,tot}}$ of about 10 minutes.

When the activity coefficient for the condensing species in the shell is reduced to $\gamma_{\text{shell}} = 10^{-3}$ to represent the partitioning
of highly miscible species to the particle phase, three orders of magnitude more mass resides in the shell than in the core at equilibrium as expected. As such, there is so much more of the condensing species in the shell, resulting in the equilibration timescale for the whole particle to be the same as that in the shell, approximately 7 minutes. The shell overshoots its equilibrium value slightly and once a time is reached where Eq. 1 is met, every subsequent time step also meets the inequality. Note that much longer than 7 minutes is required for the concentration gradients to dissolve. These concentration gradients dissipate and
the core equilibrates in about 2.5 months. This is because not only bulk diffusion in the shell is slow, but the exchange between the shell and core phases is also slow as the condensing species is much more miscible in the shell phase.

When the activity coefficient for the condensing species in the shell is $\gamma_{\text{shell}} = 10^5$, representing the partitioning of non-miscible species into organics, there are 5 orders of magnitude more mass in the core than in the shell at equilibrium. Thus, in contrast to the low $\gamma_{\text{shell}}$ case, the timescale to achieve the total gas-particle equilibrium is controlled by partitioning to the
core. Because the equilibrium concentration in the shell is so low, equilibrium between the shell and gas phase is achieved very rapidly in 2 ms. Due to slow bulk diffusion within the shell, concentration gradients in the shell persist, leading to a $\tau_{\text{eq,core}}$ of 1 hr. As partitioning into the particle bulk is hindered by the high activity coefficient in the shell, a steady-state shell concentration is maintained above the equilibrium concentration. This is maintained until the whole particle has achieved its equilibrium concentration with the concentration in the core increasing to its final equilibrium value in about one hour. This
timescale is much shorter than the low $\gamma_{\text{shell}}$ case, as the exchange rate between shell and core phases are fast, as the condensing species is more miscible in the core phase.

### 3.2 Effects of non-ideality and bulk diffusivity on equilibration timescales

We explore how equilibration timescales of semi-volatile species are affected by both non-ideality and particle viscosity. Fig. 3 shows the equilibration timescales for (a) total particle ($\tau_{\text{eq,tot}}$), (b) core ($\tau_{\text{eq,core}}$), and (c) shell ($\tau_{\text{eq,shell}}$) as a function of
activity coefficient ($\gamma_{\text{shell}}$) and bulk diffusivity ($D_{\text{b,shell}}$) of the condensing specie in the shell. $\tau_{\text{eq,tot}}$ is controlled by the phase that holds the most mass at equilibrium: $\tau_{\text{eq,tot}}$ is almost identical to $\tau_{\text{eq,core}}$, when $\gamma_{\text{shell}} > 1$ while $\tau_{\text{eq,tot}}$ is very close to $\tau_{\text{eq,shell}}$ when $\gamma_{\text{shell}} < 1$. As discussed above, the relaxation timescale of concentration gradients within the particle is characterized by $\tau_{\text{eq,core}}$.

When the particle is liquid with high bulk diffusivity ($D_{\text{b,shell}} > 10^{-10}$ cm$^2$ s$^{-1}$), $\gamma_{\text{shell}}$ has little effect on the equilibration
timescale. The effect of non-ideality becomes prominent in semi-solid or solid particles with $D_{\text{b,shell}} < 10^{-14}$ cm$^2$ s$^{-1}$. When



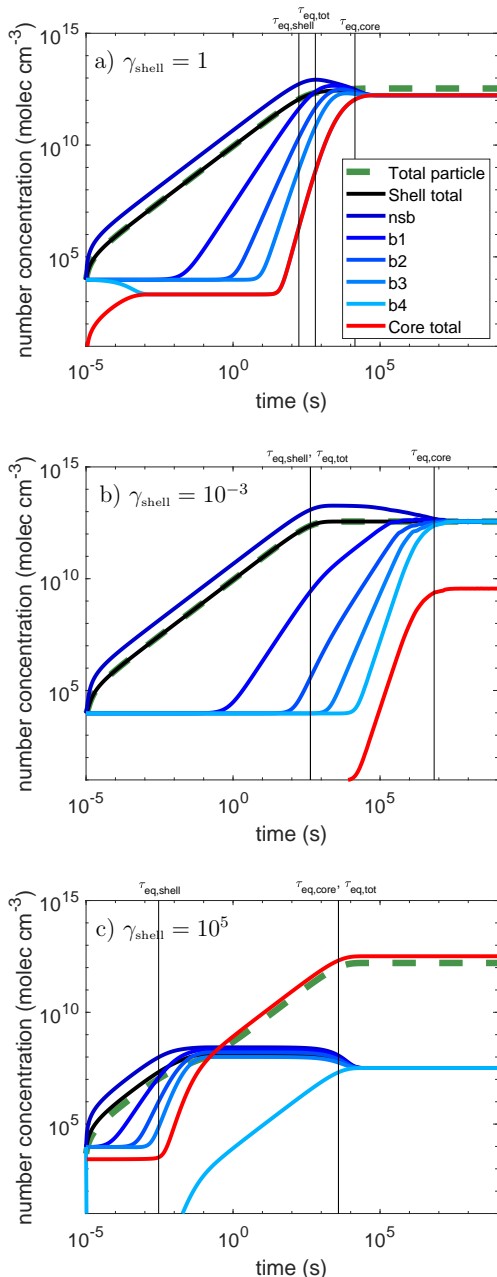

**Figure 2.** Temporal evolution of partitioning of semi-volatile species ($C^0 = 10\mu g\ m^{-3}$) into core-shell particles with the core phase to be liquid and the shell phase to be semisolid with bulk diffusivity of $D_{b,shell} = 10^{-16}\ cm^2 s^{-1}$. The activity coefficients of the condensing species in the shell phase are set to be (a) $\gamma_{shell} = 1$, (b) $\gamma_{shell} = 10^{-3}$, and (c) $\gamma_{shell} = 10^5$. The activity coefficients of the condensing species in the core phase is set to be 1 for all cases. The green dashed line is the concentration in the whole particle and the red line shows the concentration in the core. The blue lines show the concentration in each shell layer, getting progressively lighter in color from the near-surface bulk layer to the innermost shell layer (near-surface bulk, b1-4). The black line is the average concentration in the shell. The vertical black lines show the equilibrium timescales for the shell ($\tau_{eq,shell}$), core ($\tau_{eq,core}$), and total particle ($\tau_{eq,tot}$).



the shell is viscous, it takes longer for the species to move through the shell to reach the core, thus increasing $\tau_{\text{eq,core}}$. When $\gamma_{\text{shell}}$ is low, little mass needs to be transported to the core for equilibrium to be achieved and more mass needs to accumulate in the shell before a large activity gradient is achieved to move material into the core. Thus, $\tau_{\text{eq,core}}$ becomes longer when $\gamma_{\text{shell}}$ is low despite the species miscibility in the core remaining the same (i.e., $\gamma_{\text{core}} = 1$). Concentration gradients throughout the particle will be maintained until the core has reached equilibrium and thus Fig. 3b also serves to depict relaxation timescale of concentration gradients and the timescale to reach full equilibrium within particle. Comparison of $\tau_{\text{eq,core}}$ with the gas-particle equilibration timescale ($\tau_{\text{eq,tot}}$) in Fig. 3a indicates that, when $\gamma_{\text{shell}}$ is small and $D_{\text{b,shell}}$ is low, the particle may achieve equilibrium with the gas phase faster than the particle itself reaches equilibrium without concentration gradients.

The same trend is observed for $\tau_{\text{eq,shell}}$ (Fig. 3c) that non-ideality has little effect unless the shell is viscous. When the diffusion is slow, the shell can achieve its equilibrium concentration very quickly when $\gamma_{\text{shell}}$ is large, because very little mass of semivolatile species required to condense to achieve equilibrium. In contrast, when $\gamma_{\text{shell}}$ is small, much more mass is required. Since this is a closed system, once $\gamma_{\text{shell}} \leq 0.1$, more than 90 % of the available condensing species is in the shell at equilibrium and decreasing $\gamma_{\text{shell}}$ has very little impact on the equilibration timescale.

### 3.3  Volatility effects on equilibration timescale

In this section we investigate the effects of volatility of the condensing species on equilibration timescale. The same system set-up is used, while the pure compound saturation mass concentration of the condensing species ($C^0$)is varied in the range from $10^{-3}$ to $10^7$ $\mu$g m$^{-3}$ to show how the combined effects of the vapor pressure of the gas phase and non-ideal mixing impact the equilibration timescale. This range captures the volatility of typical range of atmospheric organic compounds. While there are lower volatility compounds such as highly oxygenated organic molecules (HOMs) and extremely low volatility organic compounds (ELVOCs), $C^0 = 10^{-3}$ $\mu$g m$^{-3}$ is low enough with almost no effect of activity on the equilibrium timescale of the whole particle as discussed below (Tikkanen et al., 2020; Ylisirniö et al., 2020).

Fig. 4 shows the equilibration timescale for the range of volatility cases for two different cases of diffusivity with the particle shell being liquid ($D_{\text{b,shell}} = 10^{-8}$ cm$^2$ s$^{-1}$) and semi-solid ($D_{\text{b,shell}} = 10^{-16}$ cm$^2$ s$^{-1}$). In the liquid shell (Fig. 4a), rapid equilibrium is achieved across all values of $\gamma_{\text{shell}}$ without diffusion limitations. The time to achieve equilibrium depends mainly on how much of the gas-phase species must condense to achieve equilibrium. When the volatility and the activity coefficient are higher, lower concentrations are needed to reach equilibrium; thus, the partitioning occurs very fast with short equilibration timescales. Higher concentrations are necessary with lower volatility and lower activity coefficient, leading to longer equilibration timescales.

The opposite trend is found for the semi-solid case as shown in Fig. 4b: equilibrium is reached faster when the volatility of the condensing species is lower. This is consistent with previous work on the effect of volatility on equilibration timescales of partitioning into homogeneous single phase semisolid particles in a closed system (Li and Shiraiwa, 2019). In the lower volatility cases ($C^0 < 10^{-1}$ $\mu$g m$^{-3}$), there is little effect of activity coefficient in the shell on equilibrium timescale. This is because the condensing species with sufficiently low volatility will not re-evaporate once it condenses, leading to a rapid increase in concentration at the particle surface and in the near-surface bulk. Even when the activity coefficient is high, the





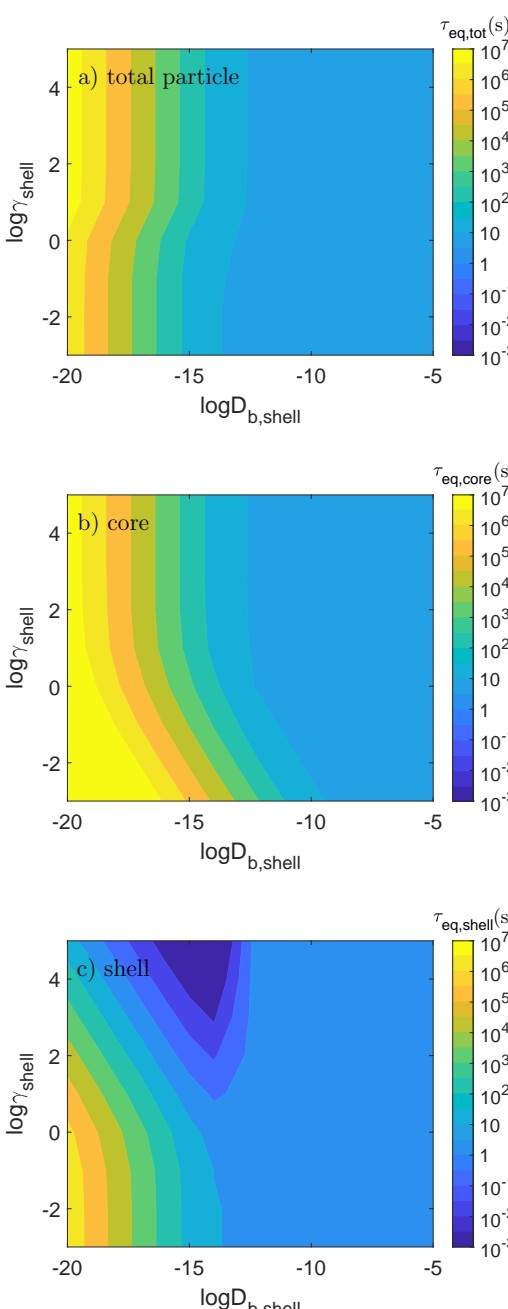

**Figure 3.** Equilibration timescales of semi-volatile species ($C^0 = 10\ \mu g\ m^{-3}$) in (a) total particle ($\tau_{eq,tot}$), (b) core ($\tau_{eq,core}$), and (c) shell ($\tau_{eq,shell}$) as a function of activity coefficient ($\gamma_{shell}$) and bulk diffusivity in the shell ($D_{b,shell}$).



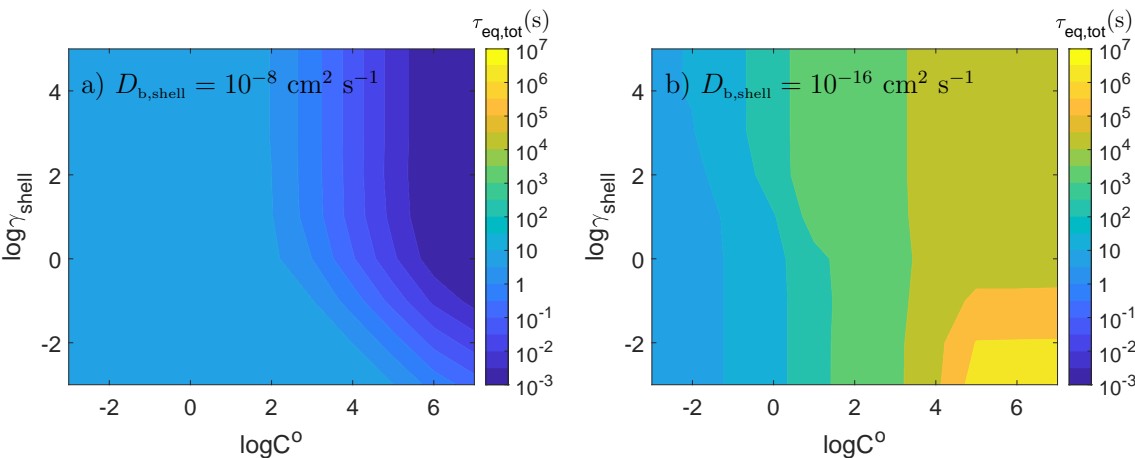

**Figure 4.** Equilibrium timescales of SOA partitioning as a function of pure compound saturation mass concentration ($C^0$) and activity coefficient of the condensing species in the shell ($\gamma_{\text{shell}}$) with bulk diffusivity of a) $D_{\text{b,shell}} = 10^{-8}$ cm$^2$ s$^{-1}$ and b) $D_{\text{b,shell}} = 10^{-16}$ cm$^2$ s$^{-1}$.

near-surface bulk will rapidly achieve a concentration that is higher than its equilibrium concentration, which then decreases upon diffusion into the particle bulk. This leads to the total particle equilibrium being achieved quickly and being dependent on the timescale for equilibrium in the shell, even if the core will contain more of the condensing species at equilibrium and the timescale to achieve full equilibrium without concentration gradients would take longer.

As the volatility increases, re-evaporation from the semisolid particle becomes significant as desorption is faster than surface-
bulk exchange and bulk diffusion in semisolid matrix, leading to an increase in equilibration timescale. For $10^{-2}$ $\mu$g m$^{-3}$ < $C^0$ < $100$ $\mu$g m$^{-3}$, we observe a small effect of activity coefficient on the total particle equilibrium: as also shown in Fig 3, $\tau_{\text{eq,tot}}$ is determined by $\tau_{\text{eq,shell}}$ when $\gamma_{\text{shell}}$ is low, while $\tau_{\text{eq,tot}}$ is controlled by $\tau_{\text{eq,core}}$ for higher $\gamma_{\text{shell}}$. The transition between these two regimes extends over $\gamma_{\text{shell}} = 1$ where the equilibrium concentrations in each phase are equal. This occurs because when the volatility is lower, more mass can be held by the shell at the same $\gamma_{\text{shell}}$ and thus $\tau_{\text{eq,shell}}$ will control $\tau_{\text{eq,tot}}$ up to higher
values of $\gamma_{\text{shell}}$.

High volatility cases ($C^0 > 10^4$ $\mu$g m$^{-3}$) show strong effects of $\gamma_{\text{shell}}$ when $\gamma_{\text{shell}}$ is low. At high volatilities, decreasing $\gamma_{\text{shell}}$ will increase the equilibrium concentration in the shell and thus the time it takes to achieve equilibrium. Overall, the equilibration timescale is long in these cases due to the rapid evaporation of these volatile species, even though less mass needs to condense to reach equilibrium.





### 3.4 Closed versus open systems

So far, all presented simulations were conducted in a closed system, where a fixed amount of the condensing species exists in the system and the gas-phase concentration decreases upon condensation. This closed system is characteristic of chamber experiments and in closed atmospheric air mass when atmospheric aerosol particles are exposed to a limited low concentration of some gas-phase species. This situation can be justified well within timescales of seconds to minutes and possibly up to hours for stagnated air mass, depending on meteorological conditions. For longer timescales, the ambient atmosphere may be better approximated as an open system due to dilution and chemical production and loss, where the gas-phase concentration remains at a constant steady-state value. Previous modeling studies that simulated SOA partitioning into ideally mixed homogeneous particles have shown that the open system can lead to longer equilibration timescales (Li and Shiraiwa, 2019; Mai et al., 2015). In this section we explore how non-ideality and phase state can impact equilibration timescales in an open system.

Fig. 5a shows the results of such simulations with $D_{b,shell} = 10^{-15}$ cm$^2$s$^{-1}$ (green line). With replenishment of the gas phase, the particle will take up more mass of the condensing species. Hence, it leads to prolongation of equilibration timescales compared to the closed system and this effect is observed across all activity coefficients in the shell. When $\gamma_{shell} > 1$, increasing $\gamma_{shell}$ has no effect on the equilibration timescale because the total mass that will condense at equilibrium is dominated by the core. In contrast, when $\gamma_{shell} < 1$, the total amount of the condensing species that can be held by the whole particle increases as $\gamma_{shell}$ decreases. Thus, $\tau_{eq,tot}$ increases sharply with decreasing $\gamma_{shell}$ in the open system, rather than staying constant as in the closed system.

Miscibility of the condensing species in the shell phase affects to what extent the equilibration timescale is increased in the open system relative to the closed system. When the condensing species is immiscible with the shell with high $\gamma_{shell}$, the open system leads to an increase in the equilibration timescale by roughly a factor of 10. For highly miscible species, for example with $\gamma_{shell} = 10^{-3}$, $\tau_{eq,tot}$ in the open system is more than four orders of magnitude higher than that in the closed system. Even with a more modest $\gamma_{shell}$ of 0.1, an open system would equilibrate at 3 hours, which is 250 times slower than the case in the closed system that equilibrates in one minute. Such big differences can lead to large discrepancies when interpreting and extrapolating closed system chamber experimental results to the atmosphere conditions.

This effect persists even when the shell is liquid in contrast to the closed system, where when the particle shell is liquid, $\gamma_{shell}$ had little to no impact on the equilibrium timescale. Fig. 5b shows $\tau_{eq,tot}$ for an open system with bulk diffusivity in the shell ranging from $D_{b,shell} = 10^{-15}$ cm$^2$ s$^{-1}$-$10^{-5}$ cm$^2$ s$^{-1}$. Even when the shell is in a liquid state, simulating an open system leads to $\tau_{eq,tot}$ 3 orders of magnitude higher than that in the closed system when $\gamma_{shell} = 10^{-3}$. Consistent with the closed system, there is little effect of diffusivity on the equilibration timescale in an open system when the particle shell is less viscous with $D_{b,shell} < 10^{-13}$ cm$^2$ s$^{-1}$.

## 4 Broader Impacts

The model simulations and analysis undoubtedly shows the complex effects and interplay of non-ideality, particle phase state, and properties of the condensing species on equilibration timescales of SOA partitioning. Experimental techniques to inves-



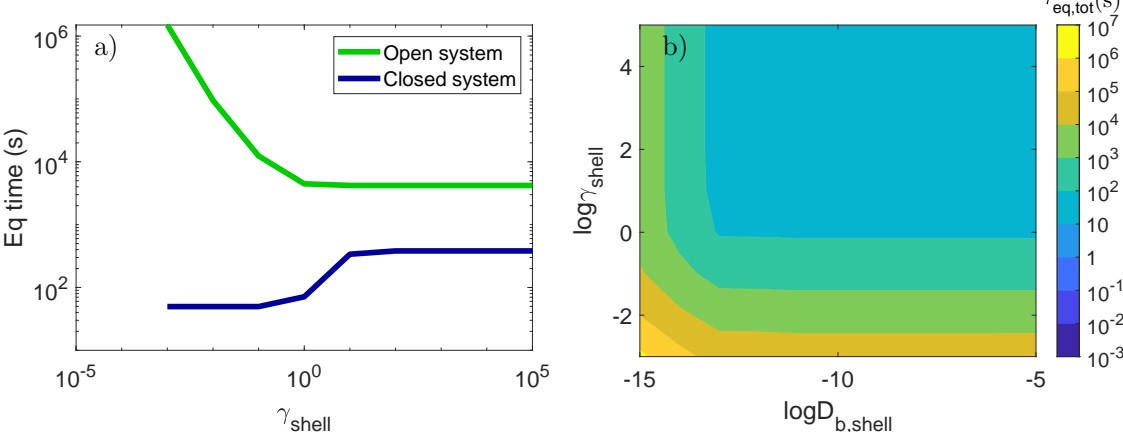

**Figure 5.** (a) Equilibrium timescale of SOA partitioning ($\tau_{\text{eq,tot}}$) with a particle with bulk diffusivity in the shell of $D_{\text{b,shell}} = 10^{-15}$ cm$^2$s$^{-1}$ and varying activity coefficients of the condensing species. The green and blue lines show the simulations in the open and closed system, respectively. (b) $\tau_{\text{eq,tot}}$ as a function of activity coefficient of the condensing species ($\gamma_{\text{shell}}$) and bulk diffusivity ($D_{\text{b,shell}}$) in the shell in the open system. Due to computational limitations, simulations using slower diffusivities than $D_{\text{b,shell}} = 10^{-15}$ cm$^2$s$^{-1}$ were not conducted.

tigate equilibration timescales rarely probe the mixing state of the aerosol population and can often only measure the total

particle composition without information on particle morphology and potential concentration gradients within the particle. In

this study we have demonstrated how concentration gradients can exist within the particle long after the particle as a whole

has achieved equilibrium with the gas phase, particularly under circumstances where the shell is able to hold large amounts

of the condensing species with low volatility or low activity coefficient in the shell. This may have impacts on heterogeneous

chemistry as under these circumstances much more of the condensing species is available at or near-surface bulk than would

be predicted assuming equilibrium partitioning.

This work can provide useful insights into some previous experimental work. Loza et al. (2013) observed that when toluene

and $\alpha$-pinene SOA are condensed sequentially on seed particles, the evaporation behavior mimics that as if only SOA from the

second precursor is present, consistent with the core-shell structure in particles (Loza et al., 2013). This then leads to minimal

evaporation when the toluene SOA coats the $\alpha$-pinene SOA, as very little evaporation of the toluene SOA was observed when

it was the only constituent in the particle. If the activity coefficient of a condensing species in the shell phase ($\gamma_{\text{shell}}$) is low

then this condition can lead to long equilibration timescales in the whole semisolid particle, as transport to the core is slow.

As suggested by Loza et al. (2013), the coating of the second precursor may be caused by non-ideal mixing effects leading to

phase separation or just very slow diffusivity and neither can be entirely ruled out. Loza et al. (2013) estimated an upper bound





on the diffusivity in toluene SOA of $D = 10^{-17}$ cm$^2$ s$^{-1}$ based on an evaporation timescale longer than 14 hours. Assuming an activity coefficient of a condensing species in toluene SOA of 1, in the simulations done here, the total particle equilibrium would be achieved in about 3 hours (Fig. 2a), but the core equilibrated much slower on the order of about a day (Fig. 2b). However, a similarly long equilibration timescale of the core can be achieved using an activity coefficient of 10 in the shell and a diffusivity of $D = 10^{-18}$ cm$^2$ s$^{-1}$ (Fig. 2b). Even with a diffusivity of $D = 10^{-17}$ cm$^2$ s$^{-1}$, equilibrium in the core would not be achieved in 14 hours with an activity coefficient of 10 in the shell (Fig. 2b).

Ye et al. (2018) observed when mixing toluene SOA and limonene SOA, that the limonene SOA initially took up a substantial amount of the toluene SOA, leading to the limonene SOA containing 15% toluene SOA almost immediately, but that was followed by much slower uptake for the next 3 hours (Ye et al., 2018). This could be evidence of the overshooting behavior as seen in this work, where the equilibrium (or steady-state) concentration is achieved rapidly as the shell layers are able to accommodate a large amount of the condensing material due to the low volatility of the condensing species and/or non-ideal mixing with low activity coefficient. If diffusivity within the particle is slow, the large concentration gradient in the particle will dissipate slowly leading to slow continuous uptake of the gas-phase until the full equilibrium is achieved.

In addition, Ye et al. (2018) performed experiments to show that even at high RH, mass exchange between SOA populations derived from toluene and $\beta$-caryophyllene SOA remained very low, despite both SOA populations containing significant fractions of semi-volatile species. In addition, $\beta$-caryophyllene SOA has been shown to have low viscosity at high RH, without inhibiting the timescale for mixing (Maclean et al., 2021). Therefore, this slow mixing could be explained by low miscibility between the two SOA populations. In this case, equilibrium may be achieved rapidly, but very little mass actually ends up mixing with the other population. As shown in (Fig. 2c), if the activity coefficient in the shell is high, the shell will achieve equilibrium rapidly even if the shell is viscous. Therefore, there was likely an activity barrier to the populations mixing at high RH as observed in their experiments.

## 5 Conclusions

We have conducted simulations to address the complex effect of non-ideal mixing and particle phase state on equilibration timescales of semivolatile species into a core-shell particle. This effect is investigated by varying the bulk diffusivity in the particle shell to cover different particle phase states from liquid, semi-solid to amorphous solid and varying activity coefficient of the condensing species in the shell. We found that the particle can achieve equilibrium with the gas phase long before the particle itself has reached full equilibrium without concentration gradients, especially when the gas phase species is preferentially miscible with the shell. If the condensing species is immiscible with the shell phase, the shell rapidly reaches quasi-equilibrium or steady state with the gas phase, while the full equilibrium in the core, the shell as a whole, and the whole particle with the gas phase is achieved much slower. If the condensing species is favorably miscible with the shell, the total particle equilibrium will reflect the equilibrium between the shell and gas phase, but the dissolution of concentration gradients in the shell and the equilibration of the core will take much longer when the shell are highly viscous with low bulk diffusivity. We have also shown that comparing these simulations to experimental work may help to reconcile apparent discrepancies between experimental



observations of fast particle-particle mixing and predictions of long mixing timescales in viscous particles based on viscosity measurements.

Large-scale models often assume both instantaneous equilibrium between a surrounding gas phase and aerosol particles which are ideally well-mixed liquids independent of the physical properties of those systems. We show that these assumptions 310 are justified under cases where the particle phase is liquid and non-ideal mixing has slight impacts on equilibration timescale between the gas phase and the particle. If the particle is viscous, condensation of miscible species in the shell can prevent mass transport to the core, prolonging equilibration timescales of full internal mixing and equilibrium of the particle with the gas phase. Hence, poor representation of particle phase state and internal mixing state can lead to overestimation of particle growth as well as heterogeneous and multiphase chemistry.

We have only considered mono-dispersed particles and a low gas-phase concentration of the condensing species so that the size of the particle remains constant throughout any simulations. Future work should consider the combined effect of phase state and non-ideality on poly-dispersed particles on the evolution of particle size distribution in atmospherically relevant concentrations and conditions.

*Data availability.* The simulation data may be obtained from the corresponding author upon request.

*Author contributions.* Both authors designed research, developed mode codes, and wrote the manuscript. M. Schervish conducted simulations.

*Competing interests.* The authors declare that they have no conflict of interest.

*Acknowledgements.* We thank U.S. National Science Foundation (AGS-1654104), U.S. Department of Energy (DE-SC0022139) and Alfred P. Sloan Foundation (MOCCIE 3, G-2020-13912) for funding.





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
