# Peer review of "Impact of phase state and non-ideal mixing on equilibration timescales of secondary organic aerosol partitioning"

_Atmospheric Chemistry and Physics, 2022_

## Author Response (AR1)

Response to Reviewer 1:

Schervish and Shiraiwa apply a multi-layer kinetic model (KM-GAP) to estimate the timescales for equilibration between the gas and particle phase when a condensable gas-phase species is present. This work expands upon previous KM-GAP studies by simulating a phase-separated, core-shell morphology and non-ideal mixing conditions in the shell. Non-ideality is explored by varying the activity coefficient of the condensing species in the shell. Other parameters are varied including the diffusion coefficient of the condensing species. The trends in the simulation outcomes are not particularly surprising and the authors do a nice job of rationalizing/ explaining the simulation results. Where these simulations are very useful is in providing some level of quantification to when diffusive limitations/ non-ideality become important for mixing timescales. This information/ data is significant in that it can help interpret experimental results and help identify the conditions that are important to account for phase separated morphologies in atmospheric chemistry models. The work is well executed and well presented, solidly within the scope of ACP, and the conclusions are well supported by the data presented. I support publication in ACP but have a few minor comments:

We appreciate Reviewer 1 for the review and very positive evaluation of our manuscript.

- Throughout the manuscript, various extrema in physical/ chemical parameters are applied to simulate a wide range of possible conditions, e.g., activity coefficients from $10^{-3}$ to $10^5$, diffusion coefficients from $10^{-5}$ to $10^{-20}$ cm$^2$/s. These values indeed cover the full range of potential values that are likely to be observed. What is missing is discussion of the relevance of the extreme low/ high values. It would have been helpful to provide some specific examples to go along with these extrema. For example, is there a specific condensing molecule that is known to have an activity coefficient of $10^{-3}$ (or $10^5$) in a organic-enriched shell? Citing specific examples would go a long way to giving the reader a sense of when these extrema could be relevant under atmospheric conditions.

This is an excellent point and additional examples of when non-idealities reach those extremes have been included. The following text has been added to Section 2.

"While many atmospherically relevant mixtures will form near-ideal solutions, with activity coefficients between 0.1 and 10, such as partitioning of similarly oxidized SOA components into an organic-rich phase extremes of miscibility can be present as well (Shiraiwa et al., 2013), For example, the thermodynamic model AIOMFAC (Aerosol Inorganic-Organic Mixtures Functional groups Activity Coefficients; Zuend et al., 2011) predicts an activity coefficient of as high as $10^8$ for α-pinene oxidation products in an aqueous electrolyte-rich phase (Shiraiwa et al., 2013) and ~$10^4$ for a strongly hydrophobic species such as hexane in a solution of hydrophilic glyoxyl and water. On the other end, AIOMFAC predicts an activity coefficient below 0.1 for small dicarboxylic acids in water-glyoxyl solutions at high water activities and similarly low activity coefficients have been experimentally determined for oxalic acid in acid-water solutions (Hyttinen et al., 2020).

- It is stated in the abstract and again in the conclusions that this work can "reconcile apparent discrepancies between experimental observations of fast particle-particle mixing and

predictions of long mixing timescales…". However, it was unclear to me exactly how the work was reconciling these discrepancies. Could the authors expand more?

There were several particle-particle mixing experiments, observing relatively fast mixing even for highly viscous particles. This is apparently contradicting with previous estimations of slow equilibration timescales for glassy or semisolid particles. Our current study provides a useful insight on apparent discrepancy, as we show that equilibration timescales can be short under certain conditions due to the interplay of non-ideality and phase state. Nevertheless, to fully resolve and explain experiments, model simulations with two different populations need to be conducted. We are currently conduct such simulations in a follow-up study. In this sense, we revised the sentence to tone down in abstract:

"Our results provide a possible explanation for discrepancies between experimental observations of fast particle-particle mixing and predictions of long mixing timescales in viscous particles and provide useful insights into description and treatment of SOA in aerosol models."

We have also clarified this issue with the following text in the section for broader impacts and conclusion as follows:

"Experimental work to probe timescales of particle-particle mixing has also been conducted by mixing two populations of particles. Ye et al (2016) showed that below 20% RH, toluene SOA does not mix appreciably, but mixes readily with a deuterated toluene SOA population at higher RH. They also showed that even at low RH, $\alpha$-pinene SOA mixes with D-toluene SOA within an hour. Ye et al (2018) extended this to include different SOA populations from isoprene, limonene, and ß-caryophyllene. While the SVOC components of isoprene and $\alpha$-pinene SOA mix rapidly in the presence of another SOA population, they showed that in some cases the properties of the other SOA population can inhibit this rapid mixing as in the case of toluene SOA mixing with limonene or ß-caryophyllene SOA. Habib and Donahue (2022) observed mixing between erythritol-coated black carbon and sugar-coated ammonium sulfate. When using a small (and presumably less viscous) sugar under high RH and temperature conditions, the erythritol achieves a steady state in the sugar-coated ammonium sulfate population in minutes. When these conditions are changed to increase the viscosity of the particles (lower RH, lower temperature, and larger sugars), equilibrium is prolonged to a few hours. Here we discuss in detail a few cases where the results of this work may help explain some findings of Ye et al (2018). However, in this work we have only simulated one particle population and thus our interpretation is limited. Future work will focus on simulating two particle populations to represent these results."

"Here we have shown that non-ideality, viscosity, and volatility all impact equilibration timescales and that combining these effects can lead to similar equilibration timescales due to their complex interplay. For example, we would expect a semisolid particle with very high viscosity would equilibrate with a gas phase slowly. In some circumstances, however, equilibrium can also be achieved rapidly for a condensing species with low $C^0$ or an immiscible species with high $\gamma$. Experimental work to probe the equilibrium timescales of semi-volatile species have focused on mixing two particle populations. Thus, to compare these results to experimental results, we must consider both the parameters affecting transport through and evaporation from one population as well as condensation and subsequent transport through a second population of aerosol, which is planned for future work."

- In Figure 2, I had a difficult time seeing the differences in the shade of blue in both the pdf and a printed version. More extreme shading difference could be helpful.

  The blue colors in Fig. 2 have been changed to be more easily distinguishable.

- Minor quibble, but from the title ("Impact of phase state…") I had anticipated more phase morphologies to be considered (e.g., a gel, partially effloresced, etc.). Since only phase separated morphologies were considered, the authors could consider a title that reflects the emphasis on phase separation.

  Following previous studies, we use "phase state" to refer to the physical state of the particle (i.e., liquid, semi-solid, or solid) rather than the morphology of the particle. Thus, we would like to keep the title as is.

Response to Reviewer 2:

Excellent study of the interplay between thermodynamic non-ideality and mass transport limitations and how it affects partitioning and, subsequently, the characteristic equilibration time for particles.

We appreciate Reviewer 2 for the review and very positive evaluation of our manuscript.

The model is well-described and the simulations in Section 3 cover the limits that are of atmospheric interest. Minor point for this Section: The presentation in Fig. 2 is a bit confusing because it is unusual. It could be improved with the addition of some illustrations that qualitatively show the differences between the three situations.

We have added text to clarify the difference in the three simulations being run with three different activity coefficients of the condensing species in the shell phase. We have also changed the colors of the blue lines (as per Reviewer 1's suggestion) to hopefully make the plots clearer.

"Three simulations are shown where the activity coefficient is changed to represent a different miscibility of the semi-volatile species in the shell phase. The activity coefficient of the condensing species in the shell phase ($\gamma_{shell}$) is set to be (a) 1 (ideal), (b) $10^{-3}$ (highly miscible), and (c) $10^5$ (hardly miscible), while that in the core phase is set to be 1 for all cases."

Although I believe the manuscript can be published as is, my only major recommendation concerns Section 4. It would be extremely helpful if there was a more comprehensive summary of existing measurements (even from laboratory-based surrogates) so that the reader can get a better idea of what cases tend to dominate equilibration times in the atmosphere. This would be valuable for future work. The current examples that are provide seem to focus on what may (or perhaps may not) be exceptional cases.

This is a great suggestion and we have added some context here into some of the mixing experiments done in Ye et al 2016, Ye et al 2018, and Habib and Donahue 2020. The following text has been to Section 4.

"Experimental work to probe timescales of particle-particle mixing has also been conducted by mixing two populations of particles. Ye et al (2016) showed that below 20% RH, toluene SOA does not mix appreciably, but mixes readily with a deuterated toluene SOA population at higher RH. They also showed that even at low RH, $\alpha$-pinene SOA mixes with D-toluene SOA within an hour. Ye et al (2018) extended this to include different SOA populations from isoprene, limonene, and ß-caryophyllene. While the SVOC components of isoprene and $\alpha$-pinene SOA mix rapidly in the presence of another SOA population, they showed that in some cases the properties of the other SOA population can inhibit this rapid mixing as in the case of toluene SOA mixing with limonene or ß-caryophyllene SOA. Habib and Donahue (2022) observed mixing between erythritol-coated black carbon and sugar-coated ammonium sulfate. When using a small (and presumably less viscous) sugar under high RH and temperature conditions, the erythritol achieves a steady state in the sugar-coated ammonium sulfate population in minutes. When these conditions are changed to increase the viscosity of the particles (lower RH, lower temperature, and larger sugars), equilibrium is prolonged to a few hours. Here we discuss in detail a few cases where the results of

this work may help explain some findings of Ye et al (2018). However, in this work we have only simulated one particle population and thus our interpretation is limited. Future work will focus on simulating two particle populations to represent these results."